# Pathological Findings in Hanging: Is the Traditional Knowledge Correct?

**DOI:** 10.3390/diagnostics14030318

**Published:** 2024-02-01

**Authors:** Graziano Domenico Luigi Crudele, Alberto Amadasi, Lorenzo Franceschetti, Cristina Cattaneo

**Affiliations:** 1LABANOF, Laboratorio di Antropologia e Odontologia Forense, Sezione di Medicina Legale, Dipartimento di Scienze Biomediche per la Salute, Università degli Studi di Milano, Via Luigi Mangiagalli 37, 20133 Milan, Italycristina.cattaneo@unimi.it (C.C.); 2Institute of Legal Medicine and Forensic Sciences, University Medical Centre Charité, University of Berlin, Turmstr. 21, Building N, 10559 Berlin, Germany

**Keywords:** hanging, asphyxia, forensic pathology, contingency table

## Abstract

Background: In forensic pathology, asphyxia presents a problem as frequently as it is thorny and challenging. Some knowledge in forensic pathology is still considered to be traditionally acquired and is not critically assessed with modern statistical or technical tools. In this study, we seek to examine the injuries that are considered to be typical of cases of hanging (neck lesions, haemorrhages, and pleural and epicardial petechiae). Methods: We evaluated whether there was, indeed, a statistically significant association between these injuries and deaths from hanging as compared to non-hanging deaths. We collected 399 cases (32 deaths by hanging and 367 cases of non-hangings), built contingency tables and performed chi-square tests for each variable (lesion) examined; we also analysed this association in various subgroups of the sample (according to sex, age and weight ranges). Results: Our results did not deviate from the expected outcome based on traditional knowledge, although they do provide a more detailed demonstration and clarification of traditional knowledge regarding this topic. Conclusions: These findings provide valuable insights for future discussion, examination and deepening of knowledge that is traditionally accepted but often undersupported in the literature.

## 1. Introduction

In forensics, asphyxial death is a frequent event, and its qualification (accidental, suicidal, homicidal or natural) is a critical question [1,2]. Among asphyxial deaths, hanging is the most frequently occurring [3]. Although the cause of death in a hanging may be quite evident, the manner of death (suicide, homicide or accident) is not always easy to define without an exhaustive and meticulous investigation [4]. This specification is necessary to completely define the diagnostic problems relating to asphyxia due to hanging; hanging is, theoretically, almost always a suicide as regards the manner of death [5]; and the other manners of death (homicide and accident) are very rare but do occur. Regarding homicide, there are two possibilities [6]: (1) a true homicidal hanging, which presupposes a disproportion of strength between the victim (much weaker or weakened) and the aggressor (much stronger or multiple aggressors); (2) a suspension of the corpse (simulation of hanging with victim dead from another cause). Typical hanging findings will obviously be different in the two cases (present in case 1, absent in case 2). Accidental cases of hanging are also quite rare and are mostly limited to autoerotic deaths, as reported in the literature [7,8]. The study of anatomo-pathological findings in hanging from a statistical perspective (which is the aim of the present study) can offer interesting cues; in the specific subset of suspensions of a corpse, the problem of this differential diagnosis is particularly acute in forensic pathology [9,10]. In other manners of death, differential diagnostics with the sole use of anatomo-pathological findings is fundamentally not possible; however, for completeness, it is relevant, presently, to mention this aspect. Considerable differences in the frequency of pathological findings are often reported in the relevant literature. Additionally, the data available for this problem are particularly numerous and often extremely variable; for example, some authors have noted a paradoxical contradiction in the extraordinary variability of laryngohyoid fractures in hanging victims [11].

By providing numbers and statistical evaluations, we aim to answer an important question for forensic pathologists regarding hanging: which diagnostic values can we give, in cases of hanging, to neck lesions, haemorrhages, and pleural and epicardial petechiae? Second, we aimed to establish whether there was any significant correlation between the findings of hanging and the physical features of the victims, a question we examined due to its recurrent appearance in the literature [12,13,14]. As far as we are aware, no study has previously attempted to numerically define these aspects using statistical methods. A real statistical comparison using contingency tables has never been carried out in forensic pathology literature, and no existing work compares hanging versus non-hanging injuries according to a classic contingency table scheme. The majority of studies available are aimed at counting the injuries within the same group of hanging cases and can be characterised as reviews [4,11,14,15,16,17,18,19,20,21,22]. Thus, any trauma of the neck (not only hanging) can cause, for example, lesions of the laryngeal skeleton [11]. The trauma of hanging, however, cannot result in any lesions of the osteocartilaginous laryngeal structures [11,13,23]. Thus, we used contingency tables to analyse this problem more precisely.

## 2. Materials and Methods

Collection of the data. We performed a retrospective study examining all autopsy cases (forensic and clinical) that came to the attention of our Institute over the course of four months (January 2016 to April 2016) and were routinely collected in the digital database of our Institute.

For documentation, we used autopsy reports in which the pathologist accurately described all the pathological findings along with their anatomical position, characteristics, dimensions and the structures involved, along with forensic photographs and schematic drawings. From the database, we retrieved information about age, sex, weight and causes of death. It should be noted that in this study, we considered only weight (not BMI), which was measured with a human dead body scale available in the dissecting room of our institution, immediately before the autopsy with the naked corpse. No cases of severe putrefaction, poor preservation of the corpse, or carbonisation occurred during the period examined. We cross-tabulated the following pathological findings (variables) with the diagnosis of asphyxial hanging: (1) fracture of the hyoid bone appreciable with palpation or dissection, (2) breaking of the horn (one or both) of the thyroid cartilage appreciable with palpation or dissection, (3) macroscopic evidence of haemorrhage in any lateral or anterior soft tissues (including bleeding at the sternocleidomastoid muscle), (4) the presence of pleural petechiae, and (5) the presence of epicardial petechiae. As is known, other anatomopathological signs have been described in the neck regarding hanging (e.g., Amussat sign or bleeding between thyroid and cricoid cartilages); however, they are not very frequent. We opted not to consider signs that occurred at a substantially low frequency (like the Amussat sign), which would certainly have caused problems in managing and applying statistical methodology.

Autoptic approach. All cases were thoroughly examined and autopsied. The examination of the neck, after the section of the skin and before any other sectorial action of the underlying muscles, provided an accurate palpatory examination. The dissection of the neck, with layered neck dissection, was performed in all cases after the dissection of the brain and before the removal of the chest and abdominal organs; during this process, attention was paid to the presence of possible haemorrhages. The palpatory examination was repeated on the laryngeal structures in situ once all the muscular planes were removed. In the following phases of the dissection, the visceral cervicothoracic block was removed, and a palpatory examination and inspection of the osteocartilaginous structures of the neck were repeated on it gradually as the surrounding soft tissues were dissected away.

Statistical Analysis. Continuous data were reported as median and interquartile range (IQR) and compared using the Mann–Whitney test. Categorical variables were reported as frequencies and percentages and analysed with the chi-square test or Fisher’s exact test, as appropriate. Mean and standard deviation, as well as median and interquartile range, were provided for continuous variables and chi-squared for categorical ones to determine a statistically significant difference; a contingency table was then created for the variables to assess whether there was association or independence. Then, to quantify this independence in terms of OR, a univariate logistic regression was performed because the individual features (age, sex, weight, height, etc.) were considered one at a time. We were unable to take into account all the features at the same time because of the small number of cases. A contingency table was built for each variable under analysis (fracture of the hyoid bone, fracture of the horn (one or both) of the thyroid cartilage, macroscopic evidence of haemorrhages in cervical soft tissues, the presence of pleural petechiae, and the presence of epicardial petechiae), as shown in Table 1.

All chi-square tests were carried out to verify the null hypothesis of independence of the variables under study, versus the alternative hypothesis of dependence. For each variable, a general test was first performed, followed by two tests dividing the sample according to gender; the samples were then tested according to weight range and then according to age group. Only tests with a *p*-value lower than the 0.05 level (indicating significance at the 0.05 level) or the 0.01 level (indicating significance at the 0.01 level) were assessed as significant; consequently, the null hypothesis was rejected for these tests. Moreover, for the significant tests, the Cramer V index, which assesses the strength of dependence’, was also produced in the output. This index varies between 0 and 1, indicating a weak dependence when it is between 0 and 0.33, an average dependence between 0.34 and 0.66, and a strong dependence between 0.67 and 1. However, problems of numerosity were found when the sample was divided according to weight and age classes.

## 3. Results

The descriptive statistical aspects of the obtained dataset are shown in the boxplots (Figure 1 and Figure 2) and in Table 2.

An overall representation of the characteristics of the non-hanging cases is provided in Table 3 and Table 4. Regarding hanging as a cause of death, our dataset included 32 deaths by hanging and 367 cases of non-hanging.

Correlation between superior thyroid horn lesions and cause of death. In the analysis of the correlation between thyroid cartilage lesions and the cause of death, the chi-square test (where numerically executable) led to rejection in all cases of the null hypothesis of independence between the two variables, with significance at the level of 0.01 in all subcategories except for the subgroup aged 30 to 40, where the test was significant at the 0.05 level. Overall, Cramer’s V index showed an appreciable intensity on the whole sample (0.457, indicating intermediate strength of association) and on each subsample (i.e., for age, sex and weight), in cases where the calculation was numerically possible. Specifically, the study of Cramer’s V index in the subsamples showed a variable strength of association between the intermediate and the intense. The only Cramer’s V index indicating a weak association was obtained from the age group between 30 and 40 years old. The subdivision into sub-samples made the test numerically impossible for the two sub-categories of weight over 80 kg (80 kg ≥ weight < 90 kg and weight ≥ 90 kg), for the sub-category under age 20, and for the two sub-categories over 70 years old. The details are shown in Table 5.

Correlation between greater hyoid horn fractures and cause of death. In the analysis of the correlation between the fractures of the greater horn of the hyoid bone and the cause of death, the chi-square test (where numerically feasible) led to rejection in all cases of the null hypothesis of independence between the two variables. The results were significant at the 0.01 level for the whole sample; depending on the sub-sample, the results were significant at either the *p* < 0.01 or *p* < 0.05 levels. Cramer’s V index overall showed an intermediate intensity across the sample (0.366). Specifically, the study of Cramer’s V index in the subsamples showed an average strength of association for all subgroups, where the calculation was numerically feasible, except for the subgroup aged 60 to 70 years old and the subgroup of female subjects, where a weak intensity of association was present. The division into sub-samples made the test numerically impossible for the three sub-categories of weight less than 70 kg, for the sub-category weighing over 90 kg, for the sub-category under the age of 20 and for the two sub-categories over 70 years. The details are shown in Table 6.

Correlation between neck soft tissue haemorrhages and cause of death. In the study of the correlation between haemorrhages of the cervical subcutaneous soft tissues and the cause of death, the chi-square test (where numerically feasible) led to rejection in all cases of the null hypothesis of independence between the two variables. The results were significant at the 0.01 level except in two cases (the subsample weighing between 80 kg and 90 kg and the subsample aged 40 to 50 years), where the results were significant at the 0.05 level. Overall, Cramer’s V index showed an intermediate intensity for the whole sample (0.462). Specifically, where the calculation was numerically executable, Cramer’s V index gave values indicating intense or intermediate association in all subsamples, except for the sub-sample of weights between 70 and 80 kg, where the value corresponded to a low intensity of association. The division into subsamples made the test numerically impossible for the sub-categories of ages under 20 and the two sub-categories of ages over 70. The details are shown in Table 7.

Correlation between subepicardial petechiae and cause of death. In the analysis of the correlation between subepicardial petechiae and the cause of death, the chi-square test (where numerically possible) led to the rejection in nearly all cases of the null hypothesis of independence between the two variables. The results were significant at two levels (0.01 or 0.05 according to the subgroup examined). Notably, the test did not allow for the rejection of the null hypothesis for the classes of weight between 80 kg and 90 kg, weight over 90 kg, or age between 30 and 40 years. The test was not numerically executable for the age group between 70 and 80 years or the age group over 80 years. Overall, Cramer’s V index results indicated a weak association for the whole group (0.323); for the various subgroups, the strength of the association varied between intermediate and weak. The details are given in Table 8.

Correlation between subpleural petechiae and cause of death. With these variables, a situation similar to the previous one occurred: in the study of the correlation between subpleural petechiae and the cause of death, the chi-square test (where numerically possible) led to the rejection of the null hypothesis of independence between the two variables, although not in all cases. The results were significant at two levels (0.01 or 0.05, depending on the subgroup examined). In particular, the test did not allow for the rejection of the null hypothesis for the weight class between 80 kg and 90 kg, for the weight class over 90 kg, for the age class under 20, or for the age class between 40 and 50 years. The test was not numerically executable for the two classes over 70 years of age. Overall, the results of Cramer’s V index indicate a weak association for the whole group (0.295); for the various subgroups, the strength of the association varied between intermediate and weak. The details are shown in Table 9.

General considerations. General considerations can be extrapolated by observing the figures. Figure 3 and Figure 4 represent the trend of Pearson’s chi-square distribution by age group and weight group, respectively.

Figure 5, Figure 6 and Figure 7 represent the trend of the Cramer’s V index separately for sex (Figure 5), age (Figure 6) and weight (Figure 7).

Generally, the subpleural and subepicardial petechiae, whose lines tend to the lower parts of the graphs, show a weaker association with the cause of death. Hyoid bone fractures, thyroid cartilage injuries, and cervical soft tissue haemorrhagic injuries are fairly firmly located in the central part of the graphs, indicating an intermediate strength of association, with intense association force peaks for thyroid cartilage injuries concerning the lowest subclasses for the weight variable and for cervical haemorrhagic lesions with the lowest weight subgroup (<50 kg) and the age subgroup between 50 and 60 years old. For some subgroups of variables, we opted not to perform any analysis due to the limited number available, deeming the lack of a result more acceptable than a result that is probably imprecise and unreliable. In non-hanging cases, the presence of traumatic neck injuries was attributed either to the trauma that caused the death or to a secondary traumatic event that occurred shortly before the death (this happens frequently among elderly people). Petechiae, on the other hand, could also be found in non-asphyxial causes of death. As mentioned above, a univariate logistic regression was performed in order to quantify independence between the variables in terms of OR. The logistic regression highlights the association between the histopathological features and the outcome of interest (whether the patient was hanging or not). For each risk factor, there was an odds ratio that accounted for the chance that each factor, individually considered, was associated (with relative *p*-value) with our outcome (hanging) (Table 10).

## 4. Discussion

Here, we present an approach using the elementary tool of contingency tables to analyse an often complex, heterogeneous and contradictory problem. A limitation of our study, however, was its small sample size and the use of a retrospective analysis, which, as observed by some authors [20,21], can limit the specificity of evaluations. The diagnostic difficulty of asphyxia stems from the lack of a real pathognomonic character of a pathognomonic sign. Consequently, it is necessary to seek diagnostic specificity with a combination of several findings [20].

To summarise our results, in the evaluation of the correlation between superior thyroid horn lesions and causes of death and the correlation between neck soft tissue haemorrhages and causes of death, the tests used here show that there is a dependence of the variables not due to the chance in all cases with variations in the level of significance in some categories (from 0.01 to 0.05). In the evaluation of the correlation between greater hyoid horn fractures and causes of death, the tests used here show that there is a dependence of the variables not due to the chance in all cases without variations in the level of significance for the entire sample (i.e., the results were significant at the 0.01 level for the whole sample).

In the evaluation of the correlation between subepicardial petechiae and cause of death, the tests used here show that there is a dependence of the variables not due to the chance in almost all cases. In particular, the test shows that the variables are independent for the classes of weight between 80 kg and 90 kg, weight over 90 kg, and age between 30 and 40 years. Additionally, in the evaluation of the correlation between subpleural petechiae and cause of death, the test shows that the variables are independent in some cases (in particular, the test shows that the variables are independent for the weight class between 80 kg and 90 kg, for the weight class over 90 kg, for the age class under 20, or for the age class between 40 and 50 years).

It should be emphasised that in our study, we did not take into consideration the possibility of artefacts, nor did we carry out a histological check, principally because we wanted to put our position in the most routine situation as possible for pathologists, who do not always have access to such methods and must rely on macroscopic evaluation only. Nonetheless, this is a limitation of this study that should be noted. Second, as can be seen from our results, sometimes there were fluctuations in the significance levels of the test, and in some cases, the test even led to accepting the null hypothesis of independence between the two variables. Unfortunately, we do not believe it is possible to provide convincing interpretative answers for the reasons why this happens, as well as because there are no studies similar to ours and we cannot make comparisons with the work of other authors. We can only now pose the question of why typical asphyxial signs of hanging in some cases (subgroups) can lose their diagnostic validity. Such an evaluation of other case studies and the repetition of the study by other authors would be interesting; after all, it was precisely the controversial nature of this topic that we wanted to find out through our study. The search for pathological findings of diagnostic value in cases of hanging has always interested forensic pathologists who have guided scientific efforts in this area. In cases of hanging, several techniques are useful for proper interpretation, including a palpatory approach, radiography, post-mortem CT, post-mortem video laryngoscopy and histological examination [24,25]. In a 2008 study, for example, external lesions (bruises, abrasions and lacerations) at the limb level were analysed to describe the usual pattern of limb lesions associated with hanging. For this purpose, two hanging and non-hanging groups were compared [26]. Another study, published in 2009, examined the significance of Simon’s haemorrhage in a case series consisting only of hangings, where a total of 178 hanging cases were observed and analysed to evaluate how many presented Simon’s sign and how many did not [27]. In 2009, a group of Canadian authors investigated the diagnostic value of cricoid cartilage lesions in relation to hanging, particularly in terms of differential diagnosis with homicide [28]. In 2018, a study by Swiss authors analysed the intervertebral vacuum phenomenon (VP). This study looked at the relationship between hanging and gas accumulations, also known as the ‘vacuum phenomenon’, which was discovered radiologically in intervertebral discs [29]. These studies are good examples of the efforts that have been made to identify new and more reliable signs that can guide forensic pathologists in diagnosis. Several studies [14,16,17,18,19,20] have found marked variability in the frequency of manifestations of cervical injuries in cases of hanging. In 2018, a study reported a variability of 0–100% [11] in the incidence of laryngohyoid fractures. Such variability can be confusing for forensic pathologists. The reasons for this variation may be ascribed to various factors, such as the lack of a common method for examining neck structures, varying degrees of thoroughness in examining neck structures, and the lack of a rigid protocol for documenting findings, among others [22]. Moreover, the interpretation of pathological findings in cases of hanging and in cases of mechanical asphyxia in general can be quite challenging. It is indisputable that the presence of laryngeal fractures (whether involving the thyroid cartilage or the hyoid bone), if it is certain that they are antemortem lesions, certainly proves the action of a traumatic force on the neck. However, this finding should be interpreted very cautiously [30]. An analogous approach is suitable for detectable haemorrhagic lesions of cervical soft tissues. The vital lesions of the laryngeal skeleton and the haemorrhagic spots on the soft tissues of the neck are pure indicators of a generic external force that has affected the neck. Therefore, using contingency tables to analyse this problem is particularly useful for shedding light on this issue via a numerical approach. Another problem is the manifestation of petechiae, which represents a typical non-specific marker traditionally related to asphyxia, and whose mechanism of formation is debated. In this regard, it is believed that a combination of various factors, such as elevated venous pressure and hypoxic injury to endothelial cells, comes into play [20]. It is common for forensic pathologists to see a greater number and distribution of petechiae in cases of asphyxia than in non-asphyxiated deaths. In our study, we sought to verify the validity of this observation for this marker (as for previous markers) and to attribute a statistical framework to it.

## 5. Conclusions

In conclusion, our results are not definitive; rather, the strength of this study, in addition to the scientific literature examined, is our attempt to demonstrate a novel method for this field and to guide future research on this type of lesion. We believe that this type of research should be continued based on the demonstrated methodological approach so that all traditional knowledge in forensic pathology can be examined using modern statistical and technical tools.

In the conclusions, we believe it is also correct to briefly repeat the limitations of our study: the first one is certainly the lack of histological closer examinations.

Another one was the low number of cases, and as it is a retrospective case-control study, the results should certainly be assessed with a critical spirit; a more balanced sample, and particularly a larger number of cases, would of course be desirable. However, this disadvantage is not believed to have a substantial impact, except for variables where the contingency tables have a zero or very low joint frequency value, which is not the case here.

In such cases, the real solidity of the statistical analysis and the results must be plainly considered with particular attention and with descriptive purposes. Apart from these limitations, there is no doubt that some conclusions are statistically supported. In this sense, we wanted to show more than results, which by their nature are perfectible, a method of proceeding which, in our opinion, must animate the evaluation of any form of knowledge, especially in forensic pathology, and, above all, of that more consolidated and ancient knowledge, which for this reason they are traditionally accepted.

To summarise the take-home messages of our study:(1)Traditional knowledge in forensic pathology must always be verified/criticised with updated approaches.(2)Our study shows that very often but not always the signs considered characteristic of hanging are associated with it with statistical significance and sometimes and in certain groups their association may depend on chance; when a statistically significant association is present, there can certainly be a variation in the strength of the association. We must therefore be very careful and prudent in considering them useful in differential diagnosis.

## Figures and Tables

**Figure 1 diagnostics-14-00318-f001:**
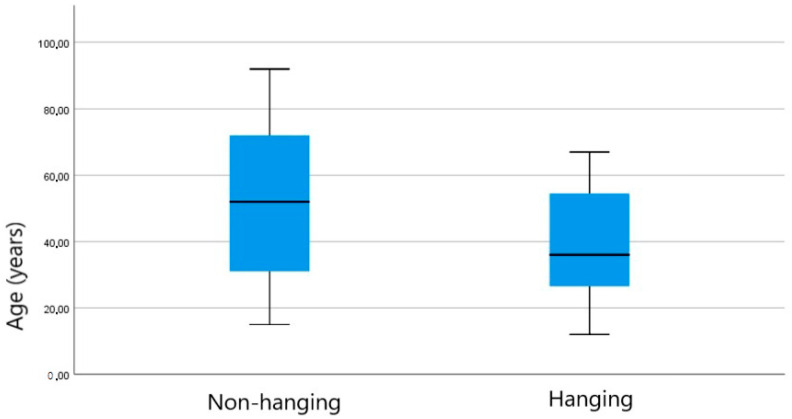
Box plot for age distribution in the two groups: hanging and non-hanging.

**Figure 2 diagnostics-14-00318-f002:**
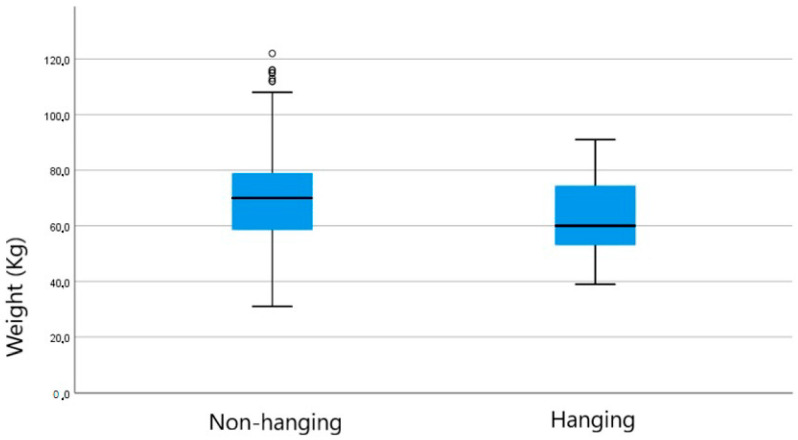
Box plot for weight distribution in the two groups: hanging and non-hanging.

**Figure 3 diagnostics-14-00318-f003:**
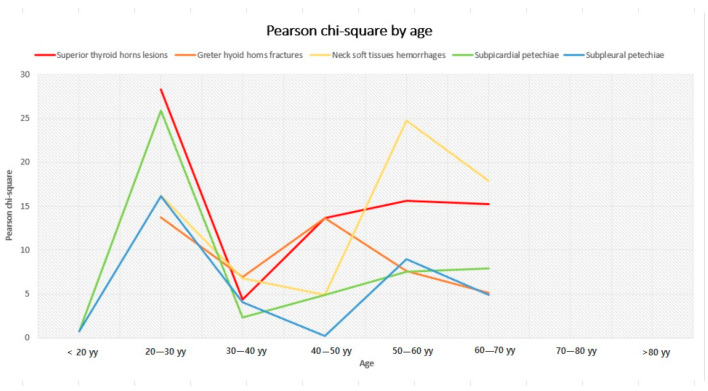
Variation of Pearson’s chi-squared test by age group.

**Figure 4 diagnostics-14-00318-f004:**
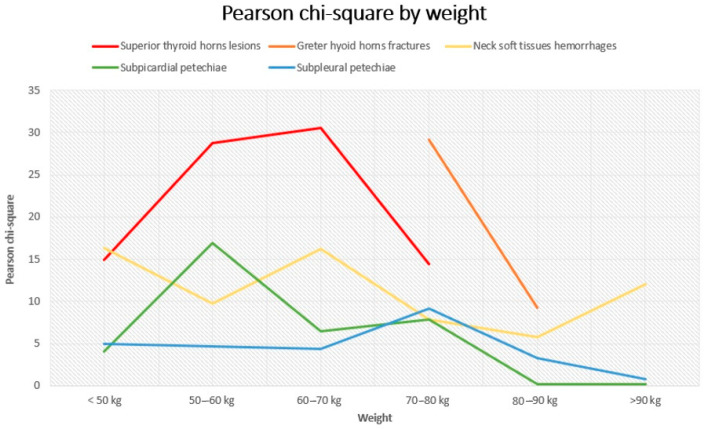
Variation of Pearson’s chi-squared by weight group.

**Figure 5 diagnostics-14-00318-f005:**
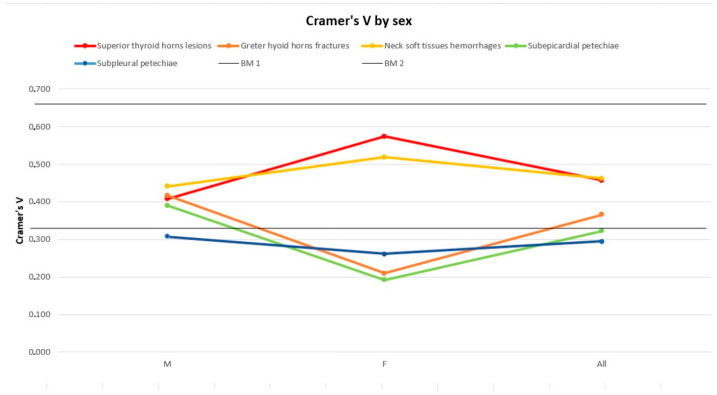
Cramér’s V distribution by sex. BM1: Benchmark 1 = 0.33; BM2: Benchmark 2 = 0.66.

**Figure 6 diagnostics-14-00318-f006:**
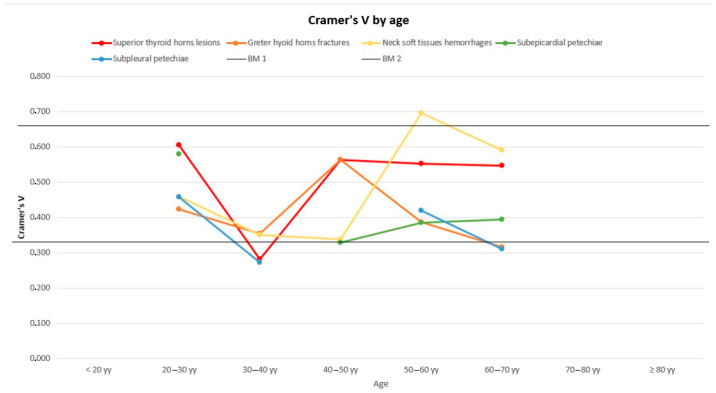
Cramér’s V distribution by age. BM1: Benchmark 1 = 0.33; BM2: Benchmark 2 = 0.66.

**Figure 7 diagnostics-14-00318-f007:**
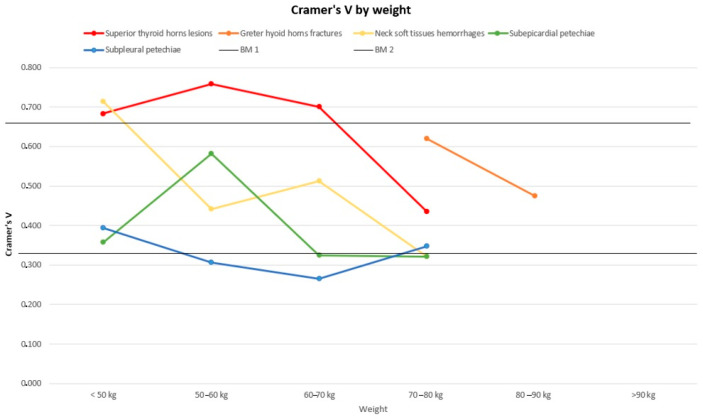
Cramér’s V distribution by weight. BM1: Benchmark 1 = 0.33; BM2: Benchmark 2 = 0.66.

**Table 1 diagnostics-14-00318-t001:** General scheme of the contingency tables used in our study.

Cause of Death	Present	Absent	Total of Row
Hanging	A	B	A + B
Non-Hanging	C	D	C + D

**Table 2 diagnostics-14-00318-t002:** Descriptive overview of the data.

		Cause of Death		
Characteristic	Overall, *N* = 399 *	Non-Hanging, *N* = 367 *	Hanging = 32 *	*p*-Value **
Age	50 (30, 70)	52 (31, 72)	36 (27, 54)	0.004
Weight	69 (57, 79)	70 (57, 79)	60 (53, 74)	0.2
Height	172 (163, 178)	172 (163, 178)	173 (164, 178)	0.9
Type of Autopsy				0.003
Forensic	316 (79%)	284 (77%)	32 (100%)	
Clinical	83 (21%)	83 (23%)	0 (0%)	
Sex				0.8
Female	120 (30%)	111 (30%)	9 (28%)	
Male	279 (70%)	256 (70)	23 (72%)	
Unknown Age	Overall = 6	Non-Hanging 6	Hanging = 0	
Unknown Weight	Overall = 113	Non-Hanging = 100	Hanging = 13	
Unknown Height	Overall = 108	Non-Hanging = 95	Hanging = 13	
Horn(s) thyroid rupture				<0.001
No	382 (96%)	361 (99%)	21 (66%)	
Yes	10 (2.5%)	3 (0.8%)	7 (22%)	
Unknown	1	1	0	
Fracture of the hyoid bone				<0.001
No	388 (97%)	363 (99%)	25 (78%)	
Yes	10 (2.5%)	3 (0.8%)	7 (22%)	
Unknown	1	1	0	
Epicardial petechiae				<0.001
No	341 (86%)	326 (89%)	15 (47%)	
Yes	57 (14%)	40 (11%)	17 (53%)	
Unknown	1	1	0	
Subpleural petechiae				<0.001
No	254 (64%)	249 (68%)	5 (16%)	
Yes	144 (36%)	117 (32%)	27 (84%)	
Unknown	1	1	0	
Cervical haemorrhagic lesions				<0.001
No	316 (79%)	311 (85%)	5 (16%)	
Yes	83 (21%)	56 (15%)	27 (84%)	

* *n* (%); Median (IQR). ** Pearson’s Chi-squared test; Wilcoxon rank sum test; Fisher’s exact test.

**Table 3 diagnostics-14-00318-t003:** Descriptive overview of the cause of death of the non-hanging cases.

Non-Hanging Cases		
Natural Causes of Death	N	%
Undetermined natural death	156	42.51%
Natural haemorrhage	10	2.72%
Myocardial infarction	10	2.72%
Pulmonary embolism	7	1.91%
Cardiac tamponade	6	1.63%
Subarachnoid haemorrhage	4	1.09%
Pneumonia	3	0.82%
Natural hemoperitoneum	3	0.82%
Multi-Organ Failure	2	0.54%
Intracerebral haemorrhage	1	0.27%
Total natural causes of death	202	55.04%
Traumatic causes of death	N	%
Multiple visceral traumatic lesions	113	30.52%
Head Trauma	33	8.99%
Burns	7	1.91%
Carbon monoxide poisoning	4	1.09%
Drowning	3	0.82%
Traumatic haemorrhage	2	0.54%
Choking	2	0.54%
Decapitation	1	0.27%
Total traumatic cause of death	165	44.96%
Total	367	100.00%

**Table 4 diagnostics-14-00318-t004:** Distribution of the manner of death in non-hanging cases.

Manner of Death in Non-Hanging Cases		
	N	%
Homicide	6	2.97%
Suicide	71	35.15%
Accident	125	61.88%
Total	202	100.00%

**Table 5 diagnostics-14-00318-t005:** Correlation between superior thyroid horn lesions and cause of death. The second and third columns represent two cells of the contingency table built for this variable.

Superior Thyroid Horn Lesions and Cause of Death				
Group	Carriers of Superior Thyroid Horn Lesions Died for Hanging Compared to the Total Hanged Cases %	*p* Value	Cramér’s V	Strength of the Association
total sample	68.8	<0.01	0.457	middle
ss: women	80	<0.01	0.574	middle
ss: men	63.6	<0.01	0.408	middle
ss: weight < 50 kg	100	<0.01	0.683	strong
ss: 50 kg ≥ weight < 60 kg	100	<0.01	0.758	strong
ss: 60 kg ≥ weight < 70 kg	100	<0.01	0.701	strong
ss: 70 kg ≥ weight < 80 kg	100	<0.01	0.435	middle
ss: 80 kg ≥ weight < 90 kg	numerically unfit for the test			
ss: weight ≥ 90 kg	numerically unfit for the test			
ss: age < 20 yy	numerically unfit for the test			
ss: 20 yy ≥ age < 30 yy	100	<0.01	0.606	middle
ss: 30 yy ≥ age < 40 yy	50	<0.05	0.282	weak
ss: 40 yy ≥ age < 50 yy	100	<0.01	0.563	middle
ss: 50 yy ≥ age < 60 yy	100	<0.01	0.553	middle
ss: 60 yy ≥ age < 70 yy	66.7	<0.01	0.547	middle
ss: 70 yy ≥ age < 80 yy	numerically unfit for the test			
ss: age ≥ 80 yy	numerically unfit for the test			

**Table 6 diagnostics-14-00318-t006:** Correlation between greater hyoid horn fractures and cause of death.

Greater Hyoid Horn Fractures and Cause of Death				
Group	Carriers of Greater Hyoid Horn Fractures Died for Hanging Compared to the Total Hanged Cases %	*p* Value	Cramér’s V	Strength of the Association
total sample	70	<0.01	0.366	middle
ss: women	50	<0.05	0.21	weak
ss: men	75	<0.01	0.417	middle
total sample	70	<0.01	0.366	middle
ss: weight < 50 kg	numerically unfit for the test			
ss: 50 kg ≥ weight < 60 kg	numerically unfit for the test			
ss: 60 kg ≥ weight < 70 kg	numerically unfit for the test			
ss: 70 kg ≥ weight < 80 kg	100	<0.01	0.62	middle
ss: 80 kg ≥ weight < 90 kg	50	<0.05	0.47	middle
ss: weight ≥ 90	numerically unfit for the test			
ss: age < 20 yy	numerically unfit for the test			
ss: 20 yy ≥ age < 30 yy	100	<0.01	0.423	middle
ss: 30 yy ≥ age < 40 yy	66.7	<0.01	0.355	middle
ss: 40 yy ≥ age < 50 yy	100	<0.01	0.563	middle
ss: 50 yy ≥ age < 60 yy	100	<0.01	0.387	middle
ss: 60 yy ≥ age < 70 yy	50	<0.01	0.317	weak
ss: 70 yy ≥ age < 80 yy	numerically unfit for the test			
ss: age ≥ 80 yy	numerically unfit for the test			

**Table 7 diagnostics-14-00318-t007:** Correlation between neck soft tissue haemorrhages and cause of death.

Neck Soft Tissue Haemorrhages and Cause of Death				
Group	Carriers of Neck Soft Tissue Haemorrhages Died for Hanging Compared to the Total Hanged Cases %	*p* Value	Cramér’s V	Strength of the Association
total sample	32.5	<0.01	0.462	middle
ss: women	36.4	<0.01	0.519	middle
ss: men	31.1	<0.01	0.441	middle
ss: weight < 50 kg	57.1	<0.01	0.714	strong
ss: 50 kg ≥ weight < 60 kg	42.8	<0.01	0.442	middle
ss: 60 kg ≥ weight < 70 kg	28.6	<0.01	0.512	middle
ss: 70 kg ≥ weight < 80 kg	25	<0.01	0.322	weak
ss: 80 kg ≥ weight < 90 kg	18.2	<0.05	0.37	middle
ss: weight ≥ 90	50	<0.01	0.69	strong
ss: age < 20 yy	numerically unfit for the test			
ss: 20 yy ≥ age < 30 yy	34.6	<0.01	0.46	middle
ss: 30 yy ≥ age < 40 yy	35.5	<0.01	0.351	middle
ss: 40 yy ≥ age < 50 yy	25	<0.05	0.338	middle
ss: 50 yy ≥ age < 60 yy	54.5	<0.01	0.696	strong
ss: 60 yy ≥ age < 70 yy	40	<0.01	0.591	middle
ss: 70 yy ≥ age < 80 yy	numerically unfit for the test			
ss: age ≥ 80 yy	numerically unfit for the test			

**Table 8 diagnostics-14-00318-t008:** Correlation between subepicardial petechiae and cause of death.

Subepicardial Petechiae and Cause of Death				
Group	Carriers of Subepicardial Petechiae Died for Hanging Compared to the Total Hanged Cases %	*p* Value	Cramér’s V	Strength of the Association
total sample	29.3	<0.01	0.323	weak
ss: women	18.2	<0.05	0.192	weak
ss: men	36.1	<0.01	0.39	middle
ss: weight < 50 kg	40	<0.05	0.358	middle
ss: 50 kg ≥ weight < 60 kg	50	<0.01	0.582	middle
ss: 60 kg ≥ weight < 70 kg	25	<0.05	0.324	weak
ss: 70 kg ≥ weight < 80 kg	25	<0.01	0.322	weak
ss: 80 kg ≥ weight < 90 kg	0	H_0_ accepted		
ss: weight ≥ 90	0	H_0_ accepted		
ss: age < 20 yy	numerically unfit for the test			
ss: 20 yy ≥ age < 30 yy	58.3	<0.01	0.58	middle
ss: 30 yy ≥ age < 40 yy	30	H_0_ accepted		
ss: 40 yy ≥ age < 50 yy	25	<0.05	0.329	weak
ss: 50 yy ≥ age < 60 yy	42.9	<0.01	0.385	middle
ss: 60 yy ≥ age < 70 yy	40	<0.01	0.394	middle
ss: 70 yy ≥ age < 80 yy	numerically unfit for the test			
ss: age ≥ 80 yy	numerically unfit for the test			

**Table 9 diagnostics-14-00318-t009:** Correlation between subpleural petechiae and cause of death.

Subpleural Petechiae and Cause of Death				
Group	Carriers of Subpleural Petechiae Died for Hanging Compared to the Total Hanged Cases %	*p* Value	Cramér’s V	Strength of the Association
total sample	18.6	<0.01	0.295	weak
ss: women	17.1	<0.01	0.262	weak
ss: men	19.2	<0.01	0.308	weak
ss: weight < 50 kg	33.3	<0.05	0.394	middle
ss: 50 kg ≥ weight < 60 kg	22.2	<0.05	0.306	weak
ss: 60 kg ≥ weight < 70 kg	10	<0.05	0.265	weak
ss: 70 kg ≥ weight < 80 kg	17.9	<0.05	0.347	middle
ss: 80 kg ≥ weight < 90 kg	12.5	H_0_ accepted		
ss: weight ≥ 90	7.1	H_0_ accepted		
ss: age < 20 yy	0	H_0_ accepted		
ss: 20 yy ≥ age < 30 yy	31.3	<0.01	0.458	middle
ss: 30 yy ≥ age < 40 yy	23.3	<0.05	0.273	weak
ss: 40 yy ≥ age < 50 yy	5	H_0_ accepted		
ss: 50 yy ≥ age < 60 yy	27.3	<0.05	0.419	middle
ss: 60 yy ≥ age < 70 yy	21.4	<0.05	0.311	weak
ss: 70 yy ≥ age < 80 yy	numerically unfit for the test			
ss: age ≥ 80 yy	numerically unfit for the test			

**Table 10 diagnostics-14-00318-t010:** Logistic regression, where the association between the histopathological features and the outcome of interest (whether the patient was hanging or not), is highlighted. For each risk factor, there is an odds ratio accounting for the chance that each factor, individually considered, is associated (with relative *p* value) with our outcome (hanging). OR = Odds Ratio, CI = Confidence Interval.

Characteristic	*N*	OR	95% CI	*p*-Value
Sex	399			
Female		—	—	
Male		1.11	0.51, 2.60	0.8
Age	393	0.97	0.95, 0.99	0.004
Weight	286	0.98	0.96, 1.01	0.2
Height	291	1.01	0.98, 1.05	0.8
Horn(s) thyroid rupture	398			
No		—	—	
Yes		37.8	12.6, 130	<0.001
Fracture of the hyoid bone	398			
No		—	—	
Yes		33.9	8.85, 165	<0.001
Epicardial petechiae	398			
No		—	—	
Yes		9.24	4.29, 20.2	<0.001
Subpleural petechiae	398			
No		—	—	
Yes		11.5	4.68, 34.6	<0.001
Cervical hemorrhagic lesions	399			
No		—	—	
Yes		30.0	12.0, 91.4	<0.001

## Data Availability

Data availability possible on motivated request to the corresponding author.

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
