# Peer review of "Pathological Findings in Hanging: Is the Traditional Knowledge Correct?"

_diagnostics, 2024, doi:10.3390/diagnostics14030318_

Round 1
Reviewer 1 Report
Comments and Suggestions for Authors
I think the paper addresses a rather interesting question, which is, are those findings typically considered as hallmarks of hanging actually solid in the forensic practice? The casuistry is wide, but I think there are some major issues. There seems to be a certain confusion between cause and manner of death. The experimental hypothesis and questions are not clearly elucidated in the introduction. The discussion is lacking, being only reported partially in the conclusions.
Introduction lines 26-30 seems a bit redundant. Also, why are we talking about suicide homicide and manner, while the rest of the paper is on hanging vs non hanging? There is a total lack of reference to the typical ways to obtain this differential diagnosis.
Introduction lines 36-37 “the basic questions for the forensic pathologist in cases of hanging: first, the value we give”. Please rephrase, weird wording
Introduction, lines 38-40 “Second, we aimed to establish if there was any significant correlation between the findings of hanging and the physical features of the victims” Is this a basic question in forensic pathology? Why would be ask ourselves something like this?
Introduction line 42 statistical comparison is repeated in close proximity
Introduction line 46 thus? Seems better something like however
Introduction, lines 47-48 “The trauma of hanging, on the other hand, could not result in any lesions of the osteocartilaginous laryngeal structures” please add a ref
Introduction, line 48 “Thus, the use of contingency tables…” I think here the statistical analysis has been confounded with the experimental design, which is not defined by the contingency tables.
Methods line 57 “From the database” not mentioned before
Methods statistical analysis lines 66-70 I think the chi square and the median and interquartile ranges are repeated. Moreover, did you check for the normality of data?
Autoptic approach, shouldn’t this be stated before statistical analysis? and is it strictly necessary? Was this dissection also performed in the clinical autopsies? I think it should be mentioned somewhere in the etxt that you included both forensic and clinical autopsies
I see caption of table 1 floating in the middle of the results and can0t read the first three lines 104-107.
Anyway, I would first say how many hanging and non-hanging case were included.
I see that type of autopsy, and age are also statistically associated to the manner of death. Could this bias the results?
Table 2, age weight and height seems rather sparse and not well identifiable. Why is the number of the unknown reported here? It looks like you did a chisquare instead of a mann-whitney in this way.
Results. I read cause of death in the text, but in the table 2 you speak about manners of death. And it goes back to cause in table 3 and 4.
Table 3, 4, 5, 6 and 7. I don’t understand the column “total died for hanging compared to the total of the group”. Please change the tables to make them clear.
Figures 1 and 2, the part A is unnecessary
Figure 3 and 4 not sure a pearson should be shown like this
Is 32 vs 367 a fair comparison? How are these 367 distributed? Are we comparing hanging with other asphyxia? Or with any cause of death?
The discussion seems more like an introduction and the conclusions are the real discussion.
Comments on the Quality of English LanguageI think the paper addresses a rather interesting question, which is, are those findings typically considered as hallmarks of hanging actually solid in the forensic practice? The casuistry is wide, but I think there are some major issues. There seems to be a certain confusion between cause and manner of death. The experimental hypothesis and questions are not clearly elucidated in the introduction. The discussion is lacking, being only reported partially in the conclusions.
Introduction lines 26-30 seems a bit redundant. Also, why are we talking about suicide homicide and manner, while the rest of the paper is on hanging vs non hanging? There is a total lack of reference to the typical ways to obtain this differential diagnosis.
Introduction lines 36-37 “the basic questions for the forensic pathologist in cases of hanging: first, the value we give”. Please rephrase, weird wording
Introduction, lines 38-40 “Second, we aimed to establish if there was any significant correlation between the findings of hanging and the physical features of the victims” Is this a basic question in forensic pathology? Why would be ask ourselves something like this?
Introduction line 42 statistical comparison is repeated in close proximity
Introduction line 46 thus? Seems better something like however
Introduction, lines 47-48 “The trauma of hanging, on the other hand, could not result in any lesions of the osteocartilaginous laryngeal structures” please add a ref
Introduction, line 48 “Thus, the use of contingency tables…” I think here the statistical analysis has been confounded with the experimental design, which is not defined by the contingency tables.
Methods line 57 “From the database” not mentioned before
Methods statistical analysis lines 66-70 I think the chi square and the median and interquartile ranges are repeated. Moreover, did you check for the normality of data?
Autoptic approach, shouldn’t this be stated before statistical analysis? and is it strictly necessary? Was this dissection also performed in the clinical autopsies? I think it should be mentioned somewhere in the etxt that you included both forensic and clinical autopsies
I see caption of table 1 floating in the middle of the results and can0t read the first three lines 104-107.
Anyway, I would first say how many hanging and non-hanging case were included.
I see that type of autopsy, and age are also statistically associated to the manner of death. Could this bias the results?
Table 2, age weight and height seems rather sparse and not well identifiable. Why is the number of the unknown reported here? It looks like you did a chisquare instead of a mann-whitney in this way.
Results. I read cause of death in the text, but in the table 2 you speak about manners of death. And it goes back to cause in table 3 and 4.
Table 3, 4, 5, 6 and 7. I don’t understand the column “total died for hanging compared to the total of the group”. Please change the tables to make them clear.
Figures 1 and 2, the part A is unnecessary
Figure 3 and 4 not sure a pearson should be shown like this
Is 32 vs 367 a fair comparison? How are these 367 distributed? Are we comparing hanging with other asphyxia? Or with any cause of death?
The discussion seems more like an introduction and the conclusions are the real discussion.
Author Response
Please see the attachment.
Revised text is in red colour

Reviewer 2 Report
Comments and Suggestions for Authors
Dear Authors!
I found your article interesting. Due to the low number of cases, it can mostly be appreciated as an introduction to a methodological approach to analyzing hanging-related findings. Also, the main difficulties arise not during differentiating hanging from non-asphyxia cases but rather from another form of strangulation (manual, ligature) - that would be more interesting. However, I think it is essential to place our forensic pathological knowledge on objective data, like this statistical approach. Therefore, I support the publication of the article. However, the following issues have to be addressed.
The article does not mention the Amussat sign, the bleeding at the sternocleidomastoid muscle, or bleeding between thyroid and cricoid cartilages. These should be mentioned, and why these were not included should be explained (the findings chosen for statistical analysis seem discretionary – which is not a problem if explained).
How was the weight of the body determined (measured)?
The weight was compared only in given weight groups. It would be more beneficial to make a statistical analysis with it as a numerical variable (as it is a numerical variable. Due to the low number of cases, it is hard to find a statistical test for it, but then again, it has to be explained.
In the case of laryngeal fractures, we commonly encounter artifacts. It is essential to examine the presence of bleeding surrounding the fracture (with histological verification if possible). Was this considered in the statistics?
It would be good to include information about the non-hanging cases, like causes of death (at least in groups), or presence of neck trauma in any cases (it would explain the presence of findings in non-hanging group). How was this control group chosen? Does it include all non-hanging autopsy cases from that period, or were some excluded (e.g., strangulation, other asphyxia)?
I would also recommend a short paragraph discussing (summarizing) the results in the disccussion section.
Formatting issues:
The Tables are not following the format required by MDPI Diagnostics:
- no lines should be present between the columns,
- no lines should be present between the rows (except first row),
- text in first row should be bald,
- the table caption should be found above the Table.
I would also recommend placing all tables below each other since the letters are hard to read in the present form (two tables at the same level).
Table 1 is wrongly placed (inside of a paragraph), and the table caption is inside the text, covering it partly (at least, in the generated PDF I received)
Author Response
Please see the attachment.
Revised text is in red colour.

Round 2
Reviewer 1 Report
Comments and Suggestions for Authors
Thanks, I think the paper has been ameliorated, although I still have some comments:
- It seems clear to me that you are evaluating hanging vs non hanging cases, which represent a different cause of death. It is ok to recall the differential diagnosis of hanging and suspension of the corpse, in the introduction. However, you talk about diagnostic classification of the case, "in relation to the manner of death", line 44 page 1/15, which is incorrect because true homicidal hangings are not included in the retrospective case series and the study will not help in establishing that.
- I see that there are some overlapping categories in the "natural causes of death", table 3. can you not put together, e.g. natural hemorrhage with subarachnoid and natural hemoperitoneum? And similar. Moreover, how is it possible to have such a high undetermined natural death cause? And among traumatic causes of death, what is cardio-respiratory insufficiency? Could the inclusion of such high visceral traumatic lesions bias the results?
- The "summarizing of results" from line 272 to 288 is certainly needed, but I think a true discussion, more than a statistical representation, would be valuable. I would suggest to rephrase, so that a reader can truly understand what the preliminary results of this new methodology are, avoiding the rejections of hypothesis, which seems like a repetition and is also unclear for a less statistically-experienced user.
- I would put all the limitations together and add the fact that some of the included cases had a low probability of showing these signs, e.g. natural hemorrhages or non-neck traumas;
- Conclusions: I see a lot of discussion and limitations, but only very few messages, except for the method of analysis, for a forensic pathology.
I think the message for the reader remains quite confusing: can we say that those signs are useful for the differential diagnosis in hanging vs non hanging? What about the other variables?
Author Response
Please, see the attachment.
